## COMMENT

# Picasso-server: a community-based, open-source processing framework for super-resolution data

Maximilian T. Strauss [ID] [1✉]

Super-resolution microscopy has become increasingly robust and performant over the last decade. Here, we report a server infrastructure that combines robust community-tested open-source processing algorithms with a workflow management system and database to enable production-grade processing of high-content super-resolution data.

Single-molecule localization microscopy (SMLM), such as PALM[1], STORM[2], and DNA-PAINT[3], all allow super-resolved image acquisition with nanometer resolution that can answer structural biological questions. Recently, SMLM applications have been demonstrated with unprecedented throughput, resolution, and increased multiplexing capabilities[4–7].

Despite these advancements, the application in routine and high-content super-resolution studies of several hundred to thousand samples is still limited. Performant and robust SMLM is technically demanding and requires optimally maintained instruments and excellently prepared samples, making routine measurements challenging.

Another technology, mass spectrometry-based proteomics, that is similarly technically demanding, already allows the routine measurement of large-scale studies and can serve as a proxy on what is needed for SMLM to achieve a similar throughput. Here, instrument and reagent performance can be continually assessed. This is achieved by tracking technical instrument parameters and testing the complete pipeline, e.g., via frequently measuring standardized quality control samples and reporting the measured proteome depth.

While similar analysis can be performed in SMLM, this is often not done automatically, and existing software packages typically are not tailored for large-scale analysis.

To address this challenge, we recently extended our python-based super-resolution library Picasso to Picasso-server[8]. Picasso is a popular open-source collection of tools and algorithms for analyzing SMLM data (Fig. 1a). While it was initially tailored for DNA-PAINT, it is capable of general SMLM and was extended by several community contributions.

The server extension has a web module and workflow management system that can be used as the building block for large-scale SMLM that are continuously processed. Direct integration to a database allows keeping track of instrument performance as well as troubleshooting individual samples.

Here, we briefly highlight the main functionalities of Picasso-server.

First, Picasso-server extends the typical Picasso workflow by connecting it to a local SQL database, which allows storing analysis settings and summary information of an experiment.

After the localization step, the resulting localization list already exhibits multiple properties that can be used to assess the performance of an experiment, such as the photon count or background of a localized spot. By calculating summary statistics, i.e., taking the mean and the standard deviation, experiments can be readily compared. Picasso-server extends the basic notion of recording summary statistics by including metrics derived from post-processing algorithms, such as the drift in an image, an experimental estimate for the localization precision

[1] Novo Nordisk Foundation Center for Protein Research, University of Copenhagen, Copenhagen, Denmark. ✉email: maximilian.strauss@cpr.ku.dk

**Fig. 1 Overview of the Picasso-server functionality and workflow. a** Picasso "Render"-Interface displaying super-resolution data. **b** Server-Functionality: Metadata, summary statistics, and metrics derived from postprocessing algorithms are added to the database after processing. **c** Left side: Web interface for Picasso-server. The database and connected raw data can be explored via multiple tabs; here, shown is the History tab. Right side: Example plots for visualizing instrument performance in the history tab. The upper panel shows Boxplots of the NeNA estimate for localization precision per day; the lower panel shows the trendline in the number of collected Photons over time. **d** Example multi-user, multi-microscope setup. Data can be collected from multiple microscopes and is transferred to a processing PC, where files are automatically processed and indexed in the database. Multiple users can access Picasso-server to track instrument performance and progress.

(i.e., with nearest neighbor analysis and the NeNA[9] value), and kinetics. While these properties are often manually calculated and inspected when examining a reconstructed image, Picasso-server chains them to the localization part in the pipeline (Fig. 1b). This allows having multiple meaningful metrics to characterize an experiment and is directly inspired by similar approaches in proteomics.

Second, Picasso-server provides a web interface to interact with the local database (Fig. 1c). Here, we rely on Streamlit, which has rapidly become one of the most popular web-hosting frameworks for Python and which we have successfully used for proteomics software already[10]. A history tab allows exploring the summary statistics and inspecting instrument performance. Here, database entries can be filtered by date and keywords and grouped. Potential deviations in experiment performance (e.g., decreasing photon numbers) become readily visible, facilitating the technical assessment and effectively scheduling maintenance cycles, identifying optimal imaging conditions or localization settings. (Fig. 1c). In an additional compare tab, users can directly compare multiple experiments using the processed molecule lists. This allows advanced comparisons such as investigating the localizations per time or comparing distributions. The compare tab is meant to simplify troubleshooting individual experiments and, e.g., identifying differences compared to a reference run. To further facilitate bookkeeping of recorded experiments, Picasso-server has a Preview tab that allows rendering of the single-molecule localizations directly within the browser.

Third, Picasso-server incorporates a file watcher that can automatically detect new files and process them with preselected settings. This includes functionality to call other Picasso functions or completely custom commands like Notifications to build tailored automation pipelines. The intended use case of the watcher relates to a lab environment with multiple instruments that continuously produce data and a central processing computer to which files are transferred (Fig. 1d). Users can therefore monitor existing experiments directly via the web server, making the access, e.g., via Remote Desktop, obsolete. Here, they can conveniently monitor the files that are currently being processed and assure performance.

Picasso-server extends the community-tested algorithms of Picasso and makes them accessible from a server interface. By connecting summary statistics and additionally quality metrics as part of a default analysis workflow, users can effectively conduct and monitor large-scale experiments. Ultimately, we hope that Picasso-server enables the next generation of high-throughput super-resolution studies. Further extensions to Picasso-server could include notification systems, automated performance warnings, or tailored recommendations on how to improve imaging quality. Like Picasso, Picasso-server is readily available as open-source with a one-click installer and a permissive MIT license.

**Reporting summary**. Further information on research design is available in the Nature Research Reporting Summary linked to this article.

## Data availability

Sample data used in the accompanying demonstration video was taken from the Single Molecule Localization Microscopy Challenge 2016, https://srm.epfl.ch.

## Code availability

The source code can be found at https://github.com/jungmannlab/picasso, which contains current releases and a link to the documentation. The Picasso package is also available in the Python Package Index. A standalone version for Windows is available on the release page. We additionally provide a Dockerfile for cross-platform support or deployment in a cloud environment. Detailed links to the individual code parts and installation instructions for Picasso-server are included in Supplementary Note 1. The documentation is additionally available as Supplementary Note 2.

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

## Acknowledgements

We thank Florian Schueder for extensive testing of the software. We thank Matthias Mann for proofreading the manuscript. MTS is supported financially by the Novo Nordisk Foundation (Grant agreement NNF14CC0001). Fig. 1d was created with BioRender.com.

## Author contributions

M.T.S. developed the software and wrote the paper.

## Competing interests

The author declares no competing interests.
