## [Peer Review File · Communications Biology]

Reviewers' comments:

Reviewer #1 (Remarks to the Author):

We recommend the publication of the article "Picasso-Server: A community-based, open-source processing framework for super-resolution data." hence it is a great add-on to the Picasso software published in [1] which proved itself to be a valuable tool in the single-molecule localization microscopy (SMLM) community.

In this manuscript, the author captures the need for a streamlined analysis platform of high-throughput SMLM data and facilitates the assessment of localization data. The author addresses this challenge by establishing a data management tool that can be utilized through a local web interface. In summary, Picasso-Server provides the following features:

- 1) Extension to a local SQL database allowing to localize SMLM data and monitor/plot the localization parameter.
- 2) Comparison of several experiments by listing/plotting the localization parameter or statistical features of drift and precision which are chained to the localization process.
- 3) Web interface with Streamlit to interact with a local database and provide plotting capacity.
- 4) History tab: technical assessment and instrument comparison and performance.
- 5) Monitoring tool that detects newly added files and localizes them automatically with pre-set parameters.

Overall, we do recommend the publication of this manuscript with minor revisions. We think that tools which enable high-throughput analysis and pave the way to early-on data quality assessment are extremely valuable for the scientific community and to our knowledge, this tool is the first of its kind. Recognizing datasets with decreased quality could hint quickly to problems in the acquisition phase and additionally would enable users to make educated decisions on discarding raw data and therefore saving storage space.

We appreciate the short and precise style of the manuscript and were pleasantly surprised by the video tutorial. It was clear and facilitated the introduction to the tool. However, we would ask the author to showcase more extensive the "file watcher" functionality. Furthermore, it would be advantageous to add two or three test localization datasets or link to a data repository like <https://srm.epfl.ch/Datasets>, which worked fine. This way new users are able to understand the compare function reliably.

The installation process on a windows machine worked without problems, but we ask the author to add a warning that the installation needs to be initiated as the administrator. If the user is not root/administrator of the PC, which is the case in many IT regimented PCs, Picasso-Server is not functional and shows error messages as indicated in the enclosed screenshot. Here, we would like to add that installer options for other operating systems (Linux, Mac) would be advantageous. SQLite is used in this manuscript. There are many different types of data management software, such as MySQL, MongoDB, Hadoop HDFS, and PostgreSQL. We ask the author to explain why SQLite is used in this work and discuss possible extensions to other databases.

Please also discuss the functionality working with data stored on virtual servers or cloud systems like AWS, o2, or HPC.

We were wondering if the "Compare" or "Watcher" function would allow comparing the statistics of various localization parameters of the same raw data set. Ultimately the automation of the process of parameter exploration for fitting and localization would be a valuable addition. Since the comparison of the different parameters will allow the user to settle on the best possible localization settings for one experiment. In the context of this work, one could add >2 settings parameter sets for the watcher function. At least discuss the potential of this feature.

Reviewer #2 (Remarks to the Author):

In this manuscript, Strauss reports a sever infrastructure that extends the capabilities of the current software package Picasso for SMLM data analysis. The Picasso-server is capable of (i) conducting

automatic single-molecule localisation of raw SMLM data sets using pre-defined detection and analysis parameters; (ii) recording a summary of statistics from the single-molecule localization lists (e.g. mean or standard deviation of photon counts, localisation precision, etc) and posterior post-processing analysis (e.g. NeNA value); (iii) comparing parameters or distributions from different files with a graphical user interface. Ultimately, all these tasks are designed with the aim to help inform the performance of SMLM experiments.

This is an interesting approach and may prove to be a useful way to identify experiments recorded with sub-optimal conditions in high-content super-resolution data, as well as to keep track of instrument performance. However, the addition of a server module that builds on the existing capabilities of the Picasso software does not seem to meet the novelty, impact and broad-interest criteria required for publication in *Communications Biology*, and it is better suited to a more specialised journal. Ultimately, the summary statistics and comparisons proposed by the author can be readily performed with a few lines of codes by any user with basic coding skills using the list of localisations of SMLM experiments obtained with Picasso or any other available SMLM data analysis package. In fact, this reviewer considers that it would be more useful to the SMLM community if the server module would allow for a more thorough analysis of high-content SMLM data sets, for example by performing other automatic tasks from the Picasso Render module, such as: (i) RCC drift correction (or drift correction via automatic identification of fiducial markers); (ii) DBSCAN clustering analysis; (iii) summary of kinetic parameters, as well as from the Picasso Filter module (i.e. to automatically filter localisations based on pre-defined user parameters for large data sets).

Picasso Server: Point-by-point response to the referees' comments

Reviewers' comments:

Reviewer #1 (Remarks to the Author):

We recommend the publication of the article “Picasso-Server: A community-based, open-source processing framework for super-resolution data.” hence it is a great add-on to the Picasso software published in [1] which proved itself to be a valuable tool in the single-molecule localization microscopy (SMLM) community.

In this manuscript, the author captures the need for a streamlined analysis platform of high-throughput SMLM data and facilitates the assessment of localization data. The author addresses this challenge by establishing a data management tool that can be utilized through a local web interface. In summary, Picasso-Server provides the following features:

- 1) Extension to a local SQL database allowing to localize SMLM data and monitor/plot the localization parameter.
- 2) Comparison of several experiments by listing/plotting the localization parameter or statistical features of drift and precision which are chained to the localization process.
- 3) Web interface with Streamlit to interact with a local database and provide plotting capacity.
- 4) History tab: technical assessment and instrument comparison and performance.
- 5) Monitoring tool that detects newly added files and localizes them automatically with pre-set parameters.

Overall, we do recommend the publication of this manuscript with minor revisions. We think that tools which enable high-throughput analysis and pave the way to early-on data quality assessment are extremely valuable for the scientific community and to our knowledge, this tool is the first of its kind. Recognizing datasets with decreased quality could hint quickly to problems in the acquisition phase and additionally would enable users to make educated decisions on discarding raw data and therefore saving storage space.

We thank the reviewer for the appreciation of our work.

We appreciate the short and precise style of the manuscript and were pleasantly surprised by the video tutorial. It was clear and facilitated the introduction to the tool. However, we would ask the author to showcase more extensive the “file watcher” functionality. Furthermore, it would be advantageous to add two or three test localization datasets or link to a data repository like <https://srm.epfl.ch/Datasets>, which worked fine. This way new users are able to understand the compare function reliably.

We thank the reviewer for the helpful suggestion to point to a reference dataset. We now include example datasets from <https://srm.epfl.ch/Datasets> on the release page, and use it in the demonstration video to work with these reference files.

The installation process on a windows machine worked without problems, but we ask the author to add a warning that the installation needs to be initiated as the administrator. If the user is not root/administrator of the PC, which is the case in many IT regimented PCs, Picasso-Server is not functional and shows error messages as indicated in the enclosed screenshot.

We thank the reviewer for highlighting the limitations with access rights and installation. We could trace down a bug that would cause the shown error message when Picasso was not installed previously. We fixed the bug and added information to launch Picasso as administrator if the user experiences problems. We also showcase instructions in the demonstration video and the release page on possible access conflicts by windows.

Here, we would like to add that installer options for other operating systems (Linux, Mac) would be advantageous.

We agree with the authors that installer options for other operating systems would be advantageous. They are planned for a future release but are not implemented at this point. Standard installation via Python is still possible on these operating systems.

To nonetheless alleviate deployment of Picasso Server, we created a Dockerfile as an additional solution. This builds on Ubuntu 20.04 and allows cross-platform execution system, e.g. Linux, Mac or Cloud system. Within the documentation we provide instructions on how to run it with a mounted volume and port forwarding. In a test setup we had a Windows host system with Docker and the Picasso Docker container running. The Picasso Server instance was accessible via the webpage from windows and could access the files on the Windows host system.

SQLite is used in this manuscript. There are many different types of data management software, such as MySQL, MongoDB, Hadoop HDFS, and PostgreSQL. We ask the author to explain why SQLite is used in this work and discuss possible extensions to other databases.

We thank the reviewer for pointing out the choice of database. We had initial experience storing the summary statistics in individual files (i.e., YAML). This had the benefit that the files were human-readable, but querying was slow for large numbers of files.

Next, we considered MongoDB. The NoSQL architecture has the benefit that the database scheme is more flexible and would be advantageous when wanting to add additional metrics later. However, here we found that packaging everything to the one-click installer was challenging. While this worked in principle, we found that compatibility issues because of conflicting SSL certificates could arise, making it difficult to work out of the box on all systems.

We don't have extensive experience with Hadoop HDFS, so we don't feel qualified to give a suitable comment on the suitability for this.

We use the Python backend SQLAlchemy, so in principle, MySQL and PostgreSQL could be used with little code changes.

Ultimately, we decided to use SQLite as it is one of the lightweight databases. Moreover, SQLite works nicely with "DB Browser for SQLite", which is an open-source tool that allows easy exploration of the database, which tails nicely with the open-source aspect of Picasso and enables transparent access. The usage of DB Browser is now also highlighted in the demonstration video.

As for extending to other databases, we see three possibilities: First, one could write an adapter script that periodically syncs the local database with a larger database, e.g., a MongoDB in the cloud. Second, one could directly push new results to a database by calling a script after localization. This is now possible with the "custom command" option in the watcher (see documentation). Third, an experienced user could make code changes within Picasso, e.g., changing directly to MongoDB.

Please also discuss the functionality working with data stored on virtual servers or cloud systems like AWS, o2, or HPC.

We thank the reviewer for raising the important question on the suitability when using virtual servers or cloud systems. Part of the functionality of Picasso server is to directly load elements from the raw data / processed HDF files. This would require it to have access to these files via the file system.

Practically, one could implement this by storing files on a network drive and mapping the drive, e.g., within the virtual server of a cloud instance. In more detail, a solution using AWS could have the AWS CLI on the instrument computers to automatically upload files to an S3 bucket and one running EC2 instance with Picasso Server that has the bucket mounted.

We were wondering if the “Compare” or “Watcher” function would allow comparing the statistics of various localization parameters of the same raw data set. Ultimately the automation of the process of parameter exploration for fitting and localization would be a valuable addition. Since the comparison of the different parameters will allow the user to settle on the best possible localization settings for one experiment. In the context of this work, one could add >2 settings parameter sets for the watcher function. At least discuss the potential of this feature.

We agree with the reviewer that assessing optimal localization settings can provide valuable insight. Picasso Server stores the localization settings, so this information was already present in the database. We now modified the way data is displayed and now duplicate entries (e.g., the same file localized with different settings) will be shown in the History tab.

For the compare, we need all localization files to be present and rely on distinct folders. As a test, we created two folders and launched two watchers with different settings. With the limitation being that raw files were duplicated, this worked as intended and allowed making comparisons in the “Compare”-Tab. We now point to this feature in the manuscript text.

Reviewer #2 (Remarks to the Author):

In this manuscript, Strauss reports a sever infrastructure that extends the capabilities of the current software package Picasso for SMLM data analysis. The Picasso-server is capable of (i) conducting automatic single-molecule localisation of raw SMLM data sets using pre-defined detection and analysis parameters; (ii) recording a summary of statistics from the single-molecule localization lists (e.g. mean or standard deviation of photon counts, localisation precision, etc) and posterior post-processing analysis (e.g. NeNA value); (iii) comparing parameters or distributions from different files with a graphical user interface. Ultimately, all these tasks are designed with the aim to help inform the performance of SMLM experiments.

This is an interesting approach and may prove to be a useful way to identify experiments recorded with sub-optimal conditions in high-content super-resolution data, as well as to keep track of instrument performance.

We thank the reviewer for appreciating Picasso Server for its intended use case to track instrument performance.

However, the addition of a server module that builds on the existing capabilities of the Picasso software does not seem to meet the novelty, impact and broad-interest criteria required for publication in Communications Biology, and it is better suited to a more specialised journal. Ultimately, the summary statistics and comparisons proposed by the author can be readily performed with a few lines of codes by any user with basic coding skills using the list of localisations of SMLM experiments obtained with Picasso or any other available SMLM data analysis package.

We agree with the reviewer that reading summary statistics based on localization data can be readily performed with a few lines of code. We see the added value in the seamless integration within the existing workflow, the extraction of additional parameters, and the possibility to quickly analyze this data and make trends visible.

In fact, this reviewer considers that it would be more useful to the SMLM community if the server module would allow for a more thorough analysis of high-content SMLM data sets, for example by performing other automatic tasks from the Picasso Render module, such as: (i) RCC drift correction (or drift correction via automatic identification of fiducial markers); (ii) DBSCAN clustering analysis; (iii) summary of kinetic

parameters, as well as from the Picasso Filter module (i.e. to automatically filter localisations based on pre-defined user parameters for large data sets).

We further agree with the suggestions that a more thorough analysis is beneficial.

We want to emphasize that Picasso server is not only meant to be a workflow management system for task automation but also to provide quality metrics to track instrument performance in a standardized way continuously. For this, it is meant to seamlessly work with the existing localize GUI, which requires the metrics to be calculated within a reasonable timeframe, ideally much shorter than the localization time.

We, therefore, carefully evaluated which additional processing steps could serve as potential quality control metrics and could be directly executed:

- RCC drift correction (i) is feasible and is already executed as part of the quality estimation.
- DBSCAN (ii). This is a very powerful technique for performing clustering analysis. However, we would like to note that this is an algorithm with $O(n^2)$ -complexity that can take very long for dense datasets. For previous datasets, we had made the experience that the reconstruction was in the range of seconds to minutes, while the DBSCAN, if successful, would take hours and would prevent the user from getting feedback in a reasonable amount of time. We, therefore, would refrain from including this.
- Kinetic parameters (iii). We appreciate the suggestion to include kinetic parameters and modified Picasso Server to include linking of localizations and estimating bright times.

To make the distinction of fast quality metrics clearer, we now changed the checkbox within Localize from “Add to database” to a sample-quality estimation button. Pressing this performs the quality check and displays the respective values. To ensure timely execution, it is limited to a subset of a maximum of $1e6$ localizations.

To cover the thorough analysis as part of the workflow in Picasso Server, we now included the option to call custom shell commands from the watcher. This allows completely customized execution of code and scripts after localization, such as clustering or linking but also non-Picasso functionality, such as sending notifications.

Within the documentation we provided sample code on how to do the following:

- Drift correction with RCC
- Linking localizations
- DBSCAN clustering
- Sending a Notification on Slack after a file is being processed

We updated the manuscript for the respective changes and included additional documentation. We hope that these changes address the reviewers' concerns.

Reviewers' comments:

Reviewer #1 (Remarks to the Author):

We appreciate the author's responses to our suggestions and questions. The revised manuscript is exhibiting improvement with the new instruction videos and the inclusion of test data.

However, the server function could not be executed due to an error and we recommend that the author reviews the release before publication. We enclosed a document containing screenshots describing the errors.

In detail the following errors were observed:

1) Installation with windows installer: The prior describe installation error for Windows machines was fixed. However, the watcher function would not pick up a newly added file. The utilized files were obtained from <https://srm.epfl.ch/Datasets> as suggested and transformed to .tif stack as described in the demonstration video. The watcher never processed anything after it started (Figure 1 in attachment). We tested iterations of having files present in the folder or adding one new file or several new files. The files were tested with the manual localization script and were read without problems.

2) Installation on MAC: Error:

The installation happens errorless (both via pip and github) but neither the localize nor the server functions started with the command prompts 'picasso localize' and 'picasso server' accordingly (Figure 2 in attachment).

We thank the author for the effort in enabling the comparison of data possible by running parallel Picasso server "Watcher" sessions rather than duplicating the raw data. However, duplicating raw data could lead to an overwhelming amount of storage needs that could be avoided if two "Watcher" instances could observe the same datasets. Each session would point to the same folder with the raw data, reducing storage needs. Then every "Watcher" session will localize the data and create .hdf5 and .yaml files related to both the data and the specific observer. (For example, add an identifier to newly created datasets "dataA_locs_watcher1" and "dataA_locs_watcher2".) This will not only reduce storage needs but also allow for a direct comparison of different localization parameters. Alternatively, the author could implement the option to run several parameters at the same time. We can understand that the authors might want to include this into a later update of the Picasso server as this might need substantial changes in their code so we do not expect this to be implemented at the moment of publication. Whichever is the decision of the author we strongly suggest adding detailed step-by-step instructions on how multiple "Watcher" can run in the video tutorial.

Reviewer #2 (Remarks to the Author):

I am satisfied with the response of the author and I believe that the revised manuscript, documentation and most importantly the Picasso server module itself is greatly improved. I especially appreciate the option to execute custom commands. I have now, no hesitation to recommend this paper for Communications Biology.

Picasso Server:

Point-by-point response to the referees' comments

Reviewers' comments:

Reviewer #1 (Remarks to the Author):

We appreciate the author's responses to our suggestions and questions. The revised manuscript is exhibiting improvement with the new instruction videos and the inclusion of test data.

We thank the reviewer for the positive feedback related to the changes in the revised manuscript.

However, the server function could not be executed due to an error and we recommend that the author reviews the release before publication. We enclosed a document containing screenshots describing the errors.

In detail the following errors were observed:

1) Installation with windows installer: The prior describe installation error for Windows machines was fixed. However, the watcher function would not pick up a newly added file. The utilized files were obtained from <https://srm.epfl.ch/Datasets> as suggested and transformed to .tif stack as described in the demonstration video. The watcher never processed anything after it started (Figure 1 in attachment). We tested iterations of having files present in the folder or adding one new file or several new files. The files were tested with the manual localization script and were read without problems.

We thank the reviewer for testing the new watcher. We hypothesize that the file watcher might not have picked up the files because of the .tif ending, whereas Picasso only checks for ome.tif-endings. Historically, Picasso was designed to only work with ome.tif from MicroManager. Although some generic .tifs will work, there is no general support for all .tifs as they can have different conventions making it very hard to ensure compatibility. Practically, renaming the .tif files to .ome.tif should make things work.

However, we conclude that the current watcher integration can be misleading and made the following changes:

- We added information about which filetypes are supported
- We added a logging module so that the status of each watcher is logged. This will allow troubleshooting when files are not being picked up.

As we, unfortunately, don't have an error message, we can't be certain about the origin of the previously reported issue. Nonetheless, we tested the functionality on Windows and macOS with systems that were at our disposal. We additionally updated the demonstration video.

2) Installation on MAC: Error:

The installation happens errorless (both via pip and github) but neither the localize nor the server functions started with the command prompts 'picasso localize' and 'picasso server' accordingly (Figure 2 in attachment).

We thank the reviewer for pointing out that the server functions could not be started on Mac with the 'localize' and 'server' commands. We now revised how modules are imported to ensure compatibility with Mac.

In the meantime, the issue was also reported on the GitHub issue page by a user from the community, and after the updates, the user reported back that it is now working on the mac system.

We thank the author for the effort in enabling the comparison of data possible by running parallel Picasso server "Watcher" sessions rather than duplicating the raw data. However, duplicating raw data could lead to an overwhelming amount of storage needs that could be avoided if two "Watcher" instances could observe the same datasets. Each session would point to the same folder with the raw data, reducing storage needs. Then every "Watcher" session will localize the data and create .hdf5 and .yaml files related to both the data and the specific observer. (For example, add an identifier to newly created datasets "dataA_locs_watcher1" and "dataA_locs_watcher2".) This will not only reduce storage needs but also allow for a direct comparison of different localization parameters. Alternatively, the author could implement the option to run several parameters at the same time. We can understand that the authors might want to include this into a later update of the

Picasso server as this might need substantial changes in their code so we do not expect this to be implemented at the moment of publication.

Whichever is the decision of the author we strongly suggest adding detailed step-by-step instructions on how multiple “Watcher” can run in the video tutorial.

We thank the reviewer for highlighting the limitations with file duplication and the suggestion to have two watchers observing the same datasets and appreciate acknowledging the code changes that are required for this. To ensure compatibility with the suggested use case, we now completely updated the watcher module. A single watcher is now able to process the same file with multiple setting groups (up to 10) and will add respective suffixes to the processed files as proposed by the author while having only one raw file. This avoids file duplication and allows comparing file performance for different processing settings. The database now stores the raw file path as well as the path of the localized so that they can be distinguished.

The new functionality is documented in the readthedocs, and below, we provide a screenshot of how this looks in practice.

Multiple parameter groups in the watcher menu

As multiple processing groups will multiply the processing load, we advise restricting the number of watchers that run simultaneously, especially when having large files and when working on systems with limited memory. In these cases, it is recommended only to run one watcher at a time. We added a performance warning regarding this to the release notes.

Reviewer #2 (Remarks to the Author):

I am satisfied with the response of the author and I believe that the revised manuscript, documentation and most importantly the Picasso server module itself is greatly improved. I especially appreciate the option to execute custom commands. I have now, no hesitation to recommend this paper for Communications Biology.

We thank the reviewer for the appreciation of the revised manuscript.

REVIEWERS' COMMENTS:

Reviewer #1 (Remarks to the Author):

We thank the author for addressing our concerns and appreciate the updated functionality of the Watcher module with multiple parameter groups. We suggest the authors make the mac version of Picasso "localize" functional before publication. We support the publication of Picasso server in Communications Biology.